# Evaluation and Comparison of Cortisol Levels in Saliva and Hair among Dental Students

Mohammed A. AlSarhan [1,*], Reham N. AlJasser [1], Saleh AlOraini [1], Syed Rashid Habib [2], Rawan Ahmad Alayoub [3], Lulwah Tawfiq Almutib [3], Haya Dokhi Aldokhi [3] and Heyam Humood AlKhalaf [3]

1   Department of Periodontics and Community Dentistry, College of Dentistry, King Saud University, Riyadh 11545, Saudi Arabia
2   Department of Prosthetic Dental Sciences, College of Dentistry, King Saud University, Riyadh 11545, Saudi Arabia
3   Dental Interns, College of Dentistry, King Saud University, Riyadh 11545, Saudi Arabia
*   Correspondence: malsarhan@ksu.edu.sa

**Abstract:** Cortisol has frequently been used as a stress marker, and the variation in cortisol levels in saliva and hair, as well as between males and females, has been reported. This study aimed to evaluate and compare the level of cortisol variation in hair and saliva among dental students of both genders. After giving written agreement, 151 students (79 males and 72 females) participated in the study. Saliva and hair samples were collected at two time intervals with a gap of three months from the same set of participants. Saliva and hair samples were utilized to measure the levels of cortisol using an enzyme-linked immunosorbent assay (ELISA). Each participant's salivary and hair cortisol levels and demographic information, including age and gender, were recorded. To examine group comparisons, two-tailed Student's $t$-tests were used ($\alpha < 0.05$). The comparisons of salivary and hair cortisol levels showed significant difference ($p < 0.05$) at two time intervals. Genderwise comparisons of the salivary and hair cortisol's levels showed significant differences ($p < 0.05$) for male students, while, for female students, the comparisons showed non-significant differences ($p > 0.05$). Comparisons between the cortisol levels of the two specimen collections at the two different time intervals indicated substantial variations ($p = 0.000$). Results confirm the variations in the cortisol levels in the saliva and hair samples. Cortisol concentrations in hair and saliva at the two times points varied. Significant variations between the male students salivary and hair cortisol levels and non-significant differences between the cortisol levels for the female students at the two time points were found. Cortisol levels in the selected subjects' saliva and hair samples varied.

**Keywords:** cortisol; saliva; hair; salivary cortisol; hair cortisol; cortisol levels

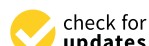



## 1. Introduction

The steroid cortisol hormone is produced and secreted by the adrenal glands. Cortisol affects several body processes, including nervous system activity, immune system performance, protein, lipid, and glucose metabolism, bone growth, and blood pressure management. It is crucial for a healthy physiology and is primarily released in response to stress [1,2]. Another chemical that our brain releases in reaction to stress and frightening situations is adrenocorticotropic hormone. Further, adrenal glands are stimulated to release cortisol and adrenaline, which results in a surge of new energy and strength [3]. Cortisol is the most common glucocorticoid in non-human primates, other animals, and humans. Although the role of cortisol in stress-related processes in both humans and animals is well-known, little is known about how well it can reflect stress levels over a prolonged period of time. Cortisol testing, which looks for increased or decreased cortisol production, is used to evaluate the pituitary and adrenal glands' functionality. Additionally, tests could be advised if a woman has irregular periods or grows more facial hair or if a child has short stature and slow growth [4,5].

Up until recently, most studies concentrated on the cortisol response from urine, serum, or saliva samples. Both saliva and serum can be used to determine the cortisol level at a specific time. Therefore, such samples can be used to test for acute changes, but both methods are highly challenging to fully evaluate prolonged systemic exposure to cortisol due to considerable daily physiological variations [6,7]. In healthy individuals, cortisol levels are highest in the morning and gradually decrease throughout the day. Because it is a noninvasive approach that generates no additional stress, research utilizing this technology shows that, although cortisol determination in saliva has indisputable advantages over plasma, its significance as a biological marker is applicable more to acute stress situations than to chronic processes [6–8].

Recently, it has been suggested that one technique to evaluate the hormonal status over an extended period is to monitor the cortisol levels in tissues like hair and nails. Hair analysis has been used for years by toxicology and forensic science to monitor exposure to exogenous substances. Because it can reveal extensive periods of time and reveal information about previous exposure to hazardous substances, hair is valued as a result [9,10]. It has advantages over body fluids, such as detection over months, noninvasive sampling, ease of storage, and strong resistance to breakdown. The introduction of hair analysis in the field of psychobiology is the result of these and other causes. Recently, cortisol analysis in hair segments has been employed in biomedical research, and it has proven to be a viable biological marker for longer periods of stress exposure than the traditional 24 h time frame [9–12].

Stress is very common among dental school students. Due to their background of high accomplishment and excellence in prior academic endeavors, as well as the fact that excellence is the standard at dentistry school, dental students have a tendency toward perfectionism, which contributes to the perception of stress [13,14]. Student performance may suffer when stress levels rise. Stress may also impair a student's ability to perform professionally by impairing their ability to pay attention, concentrate, make decisions, and develop strong patient relationships [15,16]. According to a recent study, students' bodies often experience an increase in cortisol when they suffer an academic setback, such as receiving a poor grade. A day later, it returns to normal levels for many youngsters, but for some, it remains high [17].

According to the aforementioned factors, the current study was designed with the goals of comparing and evaluating the levels of cortisol variation in saliva and hair samples taken from young adult dental students at two time intervals.

## 2. Materials and Methods

### 2.1. Study Design and Ethical Approval

The period covered by this prospective cohort research was from September 2021 to January 2022. In this study, participants from first to fifth year undergraduate dental students from King Saud University in Saudi Arabia were included. The King Saud University Medical City's institutional review board (IRB) granted its ethical approval (IRB permission # E-20-4834; Dated: 23 August 2021).

### 2.2. Sample Determination

According to G Power software, the sample size has to be at least 140, with 70 in each group, at alpha 0.05, effect size 0.5, and power 0.9. Calculations were made for descriptive statistics such mean, standard deviation (SD), and percentage.

### 2.3. Data Collection Procedure

After receiving IRB approval, the principal investigator (PI) requested the university's administration for permission to set up a classroom for study participant's recruitment. Students with aberrant skin or hair issues, baldness, a history of hormonal disorders or conditions, or who were using chronic stress or hormonal drugs were all excluded from the

study. The qualified participants provided written consent before samples were collected. Participants were given a designated reserved dental clinic to sit in.

### 2.4. Samples' Collection

Saliva and hair samples were collected at two time intervals from the same set of participants. The first collection was at the time when the students began their undergraduate courses, while the second collection was three months after the first collection.

*a.    Saliva collection*

All subjects had their saliva taken without being stimulated. Prior to the collection of saliva, participants were instructed to refrain from eating, drinking, and smoking for at least three hours. Additionally, every collection was done at a set time of the day to reduce variations in salivary production caused by the circadian rhythm. Participants were seated, and before beginning the saliva collection, participants were instructed to relax for five minutes and swallow all saliva in their mouths. They were instructed to spit into a graded test tube using a glass funnel while seated and leaning forward. After 5 min, the entire unstimulated saliva was collected, and its volume was measured [7–9,18] (Figure 1).

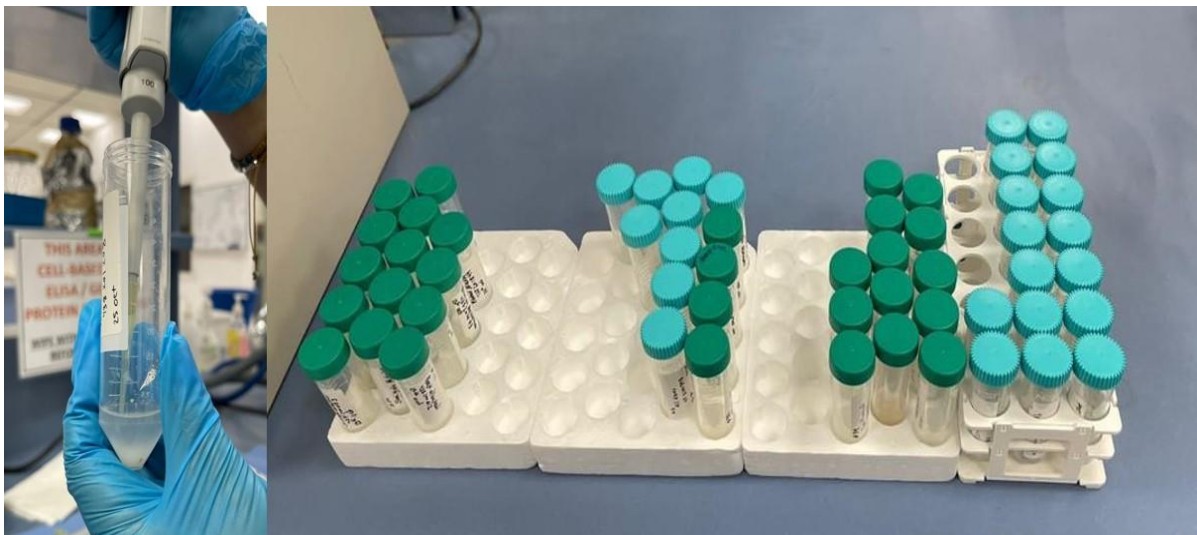

**Figure 1.** Salivary samples collection in test tubes.

*b.    Hair Samples*

Hair samples were taken as closely to the scalp as feasible from the vertex posterior region of the head. On the premise that hair grows at a pace of roughly 1 cm per month, the first 3 cm of hair from the scalp end were utilized to measure cortisol concentrations for the previous three months [10–13,19]. The hair washing and cortisol extraction methods followed a previously established protocol (Figure 2).

### 2.5. Processing and Testing of Specimens

*a.    Saliva Samples:*

Owing to the stability of cortisol, saliva samples were restored at 4–8 °C in their home refrigerator for up to 7 days before delivery to the laboratory for processing. At the laboratory, samples were either frozen at −20 °C or centrifuged to obtain a clear supernatant (the analytical sample component), then either immediately analyzed or stored at −20 °C until analysis. The samples were then thawed and re-centrifuged before analysis, ensuring that the cleanest analytical sample has been used.

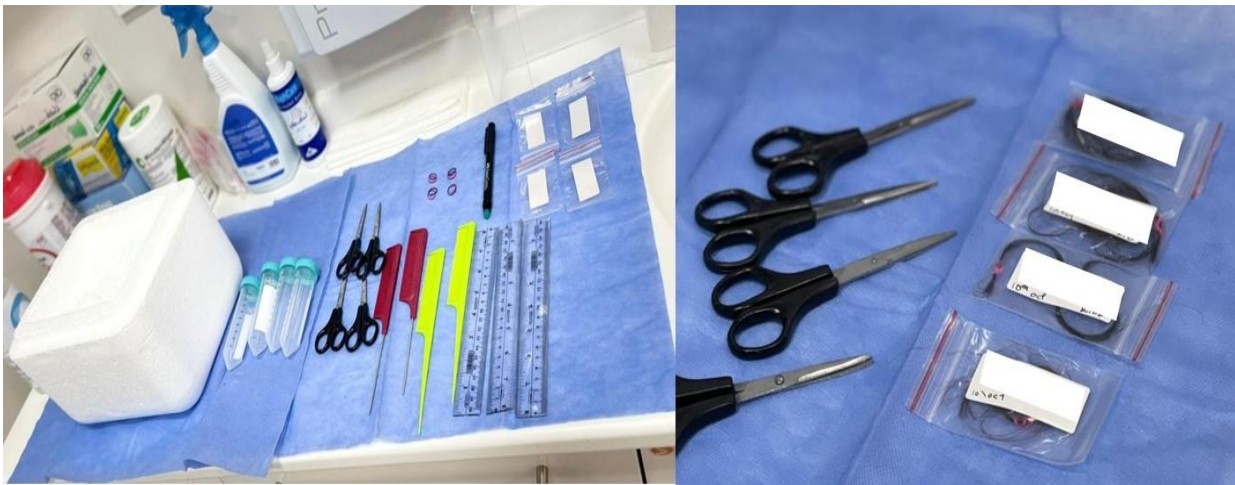

**Figure 2.** Collection tools for hair samples and some hair collections.

*b.*     *Hair Samples*

The gathered hair samples were cleaned for three minutes in a clean laminar flow hood with 2.5 mL of isopropanol in a 15 mL centrifuge tube on a tube rotator. Using a Retsch MM200 ball mill with 12 mm grinding balls and a 10 mL jar, samples were ground to a powder for 1.5 min at a speed of 25 Hz. Fisher Brand MH-14 4 decimal place balance scales were used to weigh each powdered hair sample, up to a maximum weight of 25 mg, then transferred to a 2 mL cryovial. Pure methanol (1.5 mL) was added to each sample tube to clean the hair, and the tubes were then slowly spun in an overhead rotator for 24 h. Samples were then placed into a new 2 mL cryovial and spun at 10,000 rpm for 2 min to remove the methanol using a centrifugal evaporator. The sample were finished and kept there until it was time for analysis.

Salivary cortisol buffer of pH 8 (200 L) was then added and the samples vortexed in vials before cortisol determination was carried out using a commercially available immunoassay, Stratech High Sensitivity Salivary Cortisol EIA kit (23, 24, 25).

*2.6. Enzyme-Linked Immuno-Sorbent Assays*

Saliva and hair samples were utilized to measure the levels of cortisol using an enzyme-linked immunosorbent assay (ELISA). ELISA findings were examined using a microplate reader at 450 nm (Bio-Rad Laboratories Inc., Hercules, CA, USA). The manufacturer's instructions were followed when conducting the ELISAs.17.

*2.7. Statistical Analysis*

Each participant's salivary and hair cortisol levels and demographic information, including age and gender, were recorded and calculated in Excel. For statistical analysis, the data was imported into statistical software (SPSS; IBM Corporation). To examine group comparisons, two-tailed Student's *t*-tests were used. The threshold for significance was set at $\alpha \leq 0.05$ for all statistical analyses.

**3. Results**

A total of 151 students, 79 males and 72 females, participated in the study after providing written consent. Table 1 describes the comparison of mean values of cortisol levels in saliva at two different time intervals for the participating students. The mean and standard deviation of the cortisol levels at the first and second visits were 1.54 + 0.57 and 1.30 + 0.69, respectively. The comparisons revealed significant difference ($p = 0.001$) between the cortisol levels at the two time intervals (Table 1).

**Table 1.** Cortisol levels in saliva at different time intervals for the participating students (N-151).

| Saliva Samples Collection | * Mean | Std. Deviation | Std. Error Mean | Paired Differences | ** *p*-Value |
|---|---|---|---|---|---|
| First Visit | 1.543 | 0.574 | 0.046 | 0.24 | 0.001 |
| Second Visit | 1.302 | 0.699 | 0.056 | | |
| Overall | 1.422 | 0.636 | 0.051 | | |

* Mean value was recorded in microliters; ** *p*-value was significant at $p < 0.05$.

The average cortisol levels in the saliva of the participating students at two distinct time periods are compared by gender in Figure 3. For the male subjects, the mean and standard deviation of the cortisol levels on the first and second visits were 1.79 + 0.50 and 1.28 + 0.67, respectively. For the male subjects, the comparisons showed a significant difference ($p = 0.000$) between the cortisol levels at the two time points. For the female subjects, the mean and standard deviation of the cortisol levels on the first and second visits were, respectively, 1.26 + 0.50 and 1.31 + 0.73. The comparisons showed that there was no statistically significant difference ($p = 0.585$) between the cortisol levels for the female subjects at the two time points (Figure 3).

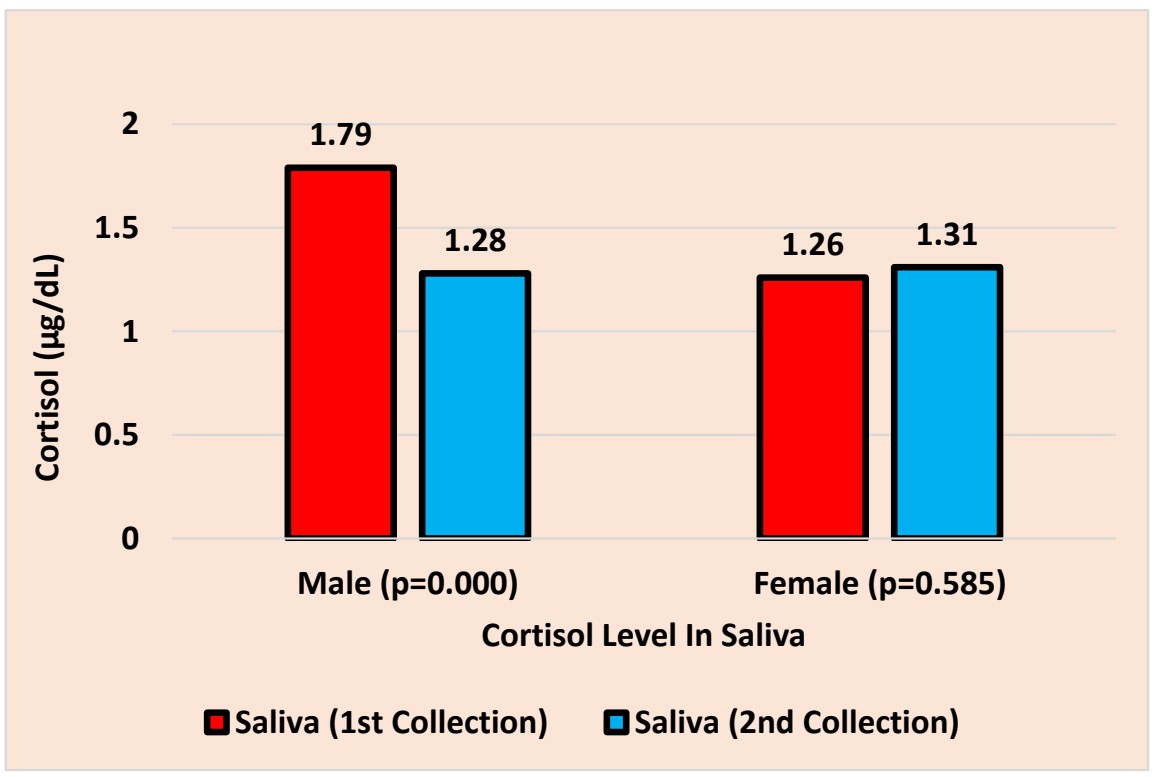

**Figure 3.** Gender wise comparison of cortisol levels in saliva for the participating students (N-151).

The comparison of the mean cortisol levels in hair samples taken from the participating students at two distinct time points is shown in Table 2. At the first and second visits, the mean and standard deviation of the cortisol levels were 1.21 + 0.78 and 0.78 + 0.61, respectively. The comparisons between the cortisol levels at the two time intervals showed a significant difference ($p = 0.000$).

**Table 2.** Cortisol levels in hair at different time intervals for the participating students (N-151).

| Hair Samples Collection | * Mean | Std. Deviation | Std. Error Mean | Paired Differences | ** *p*-Value |
|---|---|---|---|---|---|
| First Visit | 1.215 | 0.788 | 0.064 | 0.426 | 0.000 |
| Second Visit | 0.789 | 0.610 | 0.049 | | |
| Overall | 1.002 | 0.699 | 0.056 | | |

\* Mean value was recorded in microliters; ** *p*-value was significant at $p < 0.05$.

In Figure 4, the average cortisol levels in the participating students' hair samples taken at two different times are contrasted by gender. The mean and standard deviation of the cortisol levels on the first and second visits for the male participants were 1.44 + 0.79 and 0.68 + 0.57, respectively. The comparisons revealed a significant difference ($p = 0.000$) between the cortisol levels at the two time points for the males. On the first and second visits, respectively, the mean and standard deviation of the cortisol levels for the female participants were 0.96 + 0.70 and 0.90 + 0.63. The comparisons revealed that the cortisol levels for the female patients at the two time points did not differ statistically significantly ($p = 0.568$) (Figure 4).

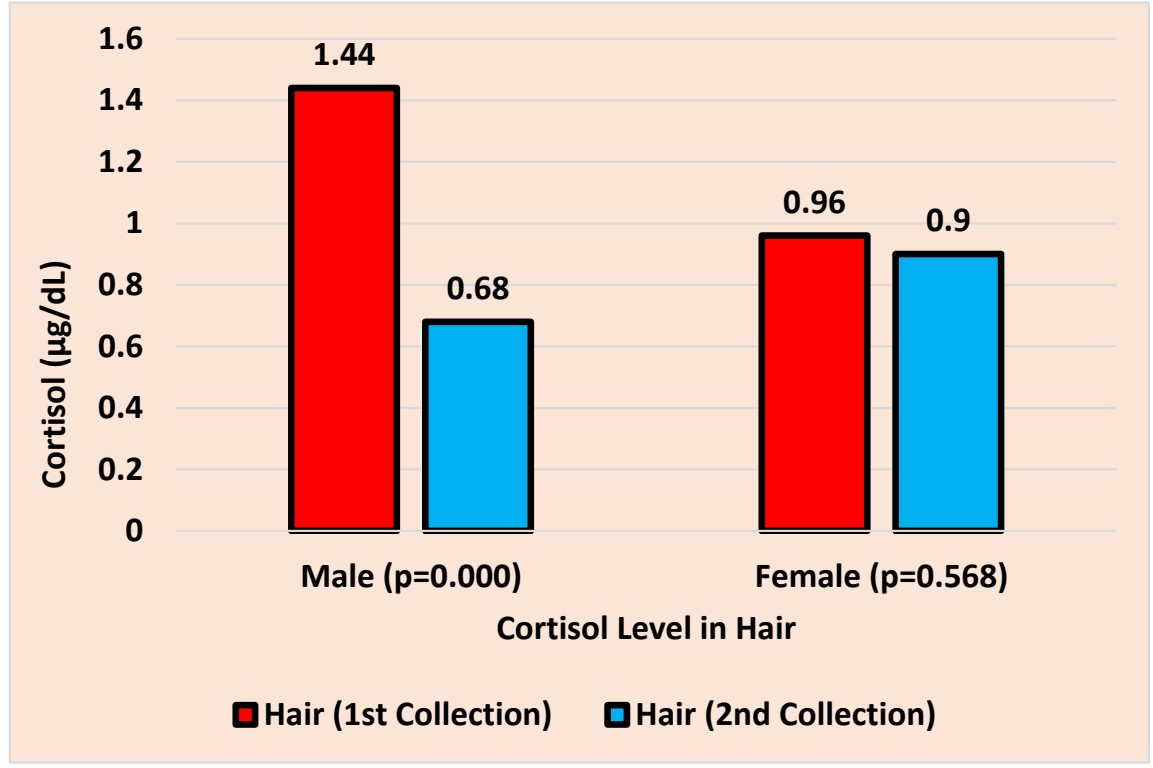

**Figure 4.** Gender wise comparison of cortisol levels in hair for the participating students (N-151).

Table 3 presents a comparison of the cortisol levels in the saliva and hair specimens of the participating subjects at the two time intervals of specimen collection. Interestingly, the comparisons between the cortisol levels of the two specimen collections at the two different time intervals indicated substantial variations ($p = 0.000$). Additional in-depth analysis of the gender comparisons of the saliva and hair cortisol levels at the two time points likewise supported the significant ($p < 0.05$) differences (Table 4). These results confirm that there are variations in the cortisol levels in the saliva and hair samples among the sample of students.

**Table 3.** Comparison of cortisol levels in saliva and hair specimens of the participating students (N-151).

| Cortisol Level | Specimens | * Mean | Std. Deviation | Std. Error Mean | Paired Differences | ** *p*-Value |
|---|---|---|---|---|---|---|
| First Visit (N = 151) | Saliva | 1.543 | 0.574 | 0.046 | 0.327 | 0.000 |
| | Hair | 1.215 | 0.788 | 0.064 | | |
| Second Visit (N = 151) | Saliva | 1.302 | 0.699 | 0.056 | 0.882 | 0.000 |
| | Hair | 0.789 | 0.610 | 0.049 | | |

* Mean value was recorded in microliters; ** *p*-value was significant at $p < 0.05$.

**Table 4.** Gender wise comparison of cortisol levels in saliva and hair specimens of the participating students at two time intervals (N-151).

| Samples Collection | Gender | Specimens | * Mean | Std. Deviation | Std. Error Mean | Paired Differences | ** *p*-Value |
|---|---|---|---|---|---|---|---|
| First Visit | Male (n = 79) | Saliva | 1.799 | 0.508 | 0.057 | 0.355 | 0.000 |
| | | Hair | 1.443 | 0.797 | 0.089 | | |
| | Female (n = 72) | Saliva | 1.262 | 0.509 | 0.060 | 0.296 | 0.003 |
| | | Hair | 0.965 | 0.702 | 0.082 | | |
| Second Visit | Male (n = 79) | Saliva | 1.287 | 0.670 | 0.075 | 0.601 | 0.000 |
| | | Hair | 0.686 | 0.572 | 0.064 | | |
| | Female (n = 72) | Saliva | 1.318 | 0.734 | 0.086 | 0.415 | 0.000 |
| | | Hair | 0.903 | 0.634 | 0.074 | | |

* Mean value was recorded in microliters; ** *p*-value was significant at $p < 0.05$.

## 4. Discussion

The stress hormone cortisol has been evaluated repeatedly in different populations, and variable cortisol levels have been found to be expressed in different body parts [1–6,18–20]. This present study aimed to assess and contrast the cortisol expressions, utilizing the saliva and hair samples of undergraduate dental students at two different time intervals. The findings of the present study showed that there were non-significant variations between the cortisol levels in the saliva and hair at the two time points for the female subjects but significant differences between the levels in the male participants' saliva and hair at the two time points. Additionally, the comparisons between the cortisol levels of the saliva specimens and the hair specimen's collections showed substantial variations in the cortisol levels.

Numerous studies have shown a significant correlation between salivary and hair cortisol concentrations [6,7,21,22]. It happens as a result of the adrenal glands' simultaneous release of cortisol into the blood and the blood's subsequent diffusion of cortisol into saliva. As a result, the blood is what determines the salivary cortisol levels at a certain time [6,7,23,24]. Cortisol expression levels in hair, on the other hand, have been documented as being moderate to weak. While cortisol concentrations in hair reflect levels of free cortisol similarly to measurements of cortisol in saliva, cortisol concentrations in hair also provide a valid long-term indication (in months) of the activity of the systemic response to chronic stresses [19,22,25]. Hair has become more intriguing as a noninvasive sample material for clinical diagnostic and research purposes due to the necessity for retrospective data regarding the technique for cortisol evaluation in human hair samples [19,22,25–27]. Although many likely mechanisms have been put out, it is still unknown how cortisol enters hair. The most widely recognized theory in this regard is cortisol diffusion from follicular capillaries into the medulla of the hair shaft during growth due to its low molecular weight and lipophilic qualities, which make it simple for the molecule to enter cells [9].

The findings of our analysis are in line with numerous studies that examined the levels of cortisol in samples of saliva and hair from male and female subjects [6,7,23–27]. Despite advancements in laboratory diagnostics, such as cortisol assays, in our opinion, it is still difficult to translate intricate the in vivo regulation of cortisol production into relatively straightforward statistical models based on laboratory tests [28]. Positive and negative feedback may depend on exogenous variables and multiple genetic polymorphisms affecting the sensitivity of the cortisol receptor, which in turn may modify its level and impact on clinical symptoms [28]. Due to the pulsing nature of its production and responsiveness to circadian cycles, cortisol levels may be affected. This could lead to misdiagnosis and a misinterpretation of laboratory findings [29]. The insufficient validation of research evaluating the connection of cortisol concentrations with other biological samples is another issue brought up by numerous other authors [27–31]. It is also critical to take into account the possibility of pre-analytical errors, especially when cortisol level assessments are more challenging. For instance, the cortisol levels in hair measurements may be impacted by poorly understood effects of chemicals used in hair care products. Chemicals like hair dye and bleach are probably used more frequently by women [32]. This might be one of the causes of the different cortisol levels seen in the hair of the male and female subjects in the current study.

It is widely established in the literature that cortisol levels vary from day to day [21,22,28,29]. The findings of the current investigation also point to the existence of these daily variations in cortisol levels. Even if there were changes in the cortisol levels for the female participants, they were not statistically significant. It is interesting to note that, for females, the variations in cortisol levels for both saliva and hair samples displayed a similar non-significant trend. In contrast, there were noticeable changes in the male participant's saliva and hair cortisol levels. These variations may be attributed to the normal variations in stress that the male participants experienced on various days of the week. In earlier investigations, it was discovered that men had higher cortisol levels than women [33]. This pattern was also shown in the results of the current investigation, wherein the levels of cortisol in the saliva and hair samples of the male participants were higher than those of the female participants. The non-significant variations in cortisol levels between the two time points for the female individuals, however, are an intriguing discovery and might make for an interesting research topic.

One of the interesting findings was the decrease in the salivary and hair cortisol levels recorded in the second visit as compared to their levels in the first visit. This downward trend of cortisol levels due to a longer period of stress exposure among subjects is also reported in the literature [34]. Young Greek adults who live in a difficult social milieu had higher perceived stress but lower cortisol levels, according to a study by Faresjö et al. [35]. These significantly lower levels of the stress hormone cortisol in the current study can be explained by changes in participants' levels of stress at the beginning of the academic year and three months later. In addition to this explanation, this study also suggests a potential biological mechanism by which the cortisol levels of the participating students may have been suppressed and downregulated as a result of the stress response mechanism. More research is necessary to determine the effects of stress caused by repeated crises among dentistry students because of cortisol expression over a longer perspective.

The study is constrained by the sparse participant pool. Despite having fewer study subjects overall than earlier studies in the same field, the subjects in the current study were all from the same professional background and the same age group and were homogeneous in terms of hormonal status. The participants' enrollment was carried out in a controlled setting with healthy volunteers. This is in contrast to some other research, wherein patients were recruited as subjects, whose readings of cortisol may have been impacted by patients with various illnesses and medications. Although the study's sample size was modest, it was very carefully chosen, and it included only people who were healthy, which is a significant advantage. The influence of different hair treatments, such as dying and coloring, among the subjects may have caused variances in the cortisol levels for the hair

specimens, and this is a potential research topic that needs to be further examined in future studies. In addition, a study of cortisol levels in association with other hormonal levels may also be explored in upcoming research.

These study's findings are only useful for understanding the variations in cortisol levels in saliva and hair samples, as well as levels in males and females at two separate times. There is no way to extrapolate these findings to people with varied social and occupational backgrounds. The association between daily variations in work-related stress and variations in cortisol levels at various times of the day has been the subject of more research requests.

## 5. Conclusions

Overall, there were noticeable variations between the levels of cortisol in hair and saliva at the two time points. However, non-significant differences between the cortisol levels for the female subjects and significant differences between the salivary and hair cortisol levels of the male subjects at the two time intervals were found. The results also revealed the existence of variations in the cortisol levels in the saliva and hair samples for the recruited participants.

**Author Contributions:** Conceptualization, M.A.A., R.N.A. and S.A.; methodology, M.A.A., R.N.A. and S.A.; software, M.A.A. and S.R.H.; validation R.N.A., S.A. and M.A.A.; formal analysis, R.N.A. and S.R.H.; investigation, R.N.A., R.A.A., L.T.A., H.D.A. and H.H.A.; resources, S.A.; data curation, S.R.H. and S.A.; writing—original draft preparation, all authors; writing—review and editing. All Authors; visualization, R.N.A.; supervision, project administration, M.A.A. All authors have read and agreed to the published version of the manuscript.

**Funding:** This research received no external funding.

**Institutional Review Board Statement:** The project was granted approval by the institutional review board (IRB), and its project activities were approved by the institutional committee of research ethics at the College of Dentistry Research Center, King Saud University, Riyadh, Saudi Arabia (IRB permission # E-20-4834). The study was conducted in accordance with the Helsinki Declaration of 1975, as revised in 2013.

**Informed Consent Statement:** Informed consent was obtained from all subjects involved in the study.

**Data Availability Statement:** Data is available on request from corresponding author.

**Acknowledgments:** Authors' are very grateful to Rhodanne Nicole A. Lambarte for her assistance in the laboratory part and Shaik S. Ahamed for the statistical analysis. Furthermore, the authors would like to thank the College of Dentistry Research Center at King Saud University, Riyadh, Saudi Arabia, for all the support provided for this study.

**Conflicts of Interest:** The authors declare no conflict of interest.

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
