# Peer review of "Evaluation and Comparison of Cortisol Levels in Saliva and Hair among Dental Students"

_applsci, doi:10.3390/app13020678_

Round 1

Reviewer 1 Report

Is an intersting approach and some nice data. Methodological section needs some extrawork and referencing. Why choosing that specific part to collect hair. No reference is given. etc.Why 5 minutes for saliva etc . Again no reference here. Some exclusion criteria are also a little but subjective. But it is a nice study, with some interst for the readers.

Statistics is very basic, but still ok.

No date of the ethical approval is given.

Introduction and Conclusions are well constructed.

Conclusions seems balanced and ok.

Author Response

Reviewer 1:

Is an intersting approach and some nice data. Methodological section needs some extrawork and referencing.

Authors’ response: The authors would like to thank the respected reviewer for reviewing and providing an insightful and constructive feedback on the manuscript that certainly helped us to improve content. The authors worked thoroughly and revised the manuscript based on the provided suggestions. Please find a point-by-point response to the comments below:

Why choosing that specific part to collect hair. No reference is given. etc.

Response: The authors would like to thank the reviewer for the comment. Hair samples were taken as closely to the scalp as feasible from the vertex posterior region of the head. On the premise that hair grows at a pace of roughly 1 cm per month. Several research studies have employed the same methodology. New references as well as old ones are now added and the references adjusted accordingly to this part of methodology. [Lines 123]

Why 5 minutes for saliva etc . Again no reference here.

Response: The authors would like to thank the reviewer for the comment. Several research studies have employed the same methodology for the salivary collection. New references as well as old ones are now added and the references adjusted accordingly to this part of methodology. [Lines 113]

Some exclusion criteria are also a little but subjective. But it is a nice study, with some interst for the readers.

Response: The authors would like to thank the reviewer for the positive feedback.

Statistics is very basic, but still ok.

Response: The authors would like to thank the reviewer for the positive feedback.

No date of the ethical approval is given.

Response: The authors would like to thank the reviewer for the comment. The date is now added to the methodology [Line 89].

Introduction and Conclusions are well constructed.

Response: The authors would like to thank the reviewer for the positive feedback.

Conclusions seems balanced and ok.

Response: The authors would like to thank the reviewer for the positive feedback.

Finally, the authors would like to thank the reviewer for the constructive feedback and helping the authors to improve the contents and quality of this manuscript. We hope the quality of the manuscript has been improved and will be acceptable for publication.

Sincerely,

Reviewer 2 Report

My comments:

1, Line 252 to 256: “…the cortisol levels in hair measurements may be impacted by poorly understood effects of chemicals used in hair care products….” “This might be one of the causes of the different cortisol levels seen in the hair of the male and female subjects in the current study.”

Did the authors collect any data that participants used hair dye when collecting their hair samples? If so, what percent of the female did dye their hair?  In this case/manuscript, do the male cortical levels in their hair more reflect the actual levels compared with female’s ?  If so, it should be discussed.

2, Line 266: “This pattern was also shown in the results of the current investigation, where the levels of cortisol in the saliva and hair samples of the male participants were higher than those of the female participants.”

But, Figure 3 and 4 show that there is no significantly differences of cortisol level in Saliva between male and female at 1st collection: Why?  Is there any significantly difference of those between male and female at 2nd collection?

Author Response

Reviewer 2:

Authors’ response: The authors would like to thank the respected reviewer for reviewing and providing an insightful and constructive feedback on the manuscript that certainly helped us to improve content. The authors worked thoroughly and revised the manuscript based on the provided suggestions. Please find a point-by-point response to the comments below:

1, Line 252 to 256: “…the cortisol levels in hair measurements may be impacted by poorly understood effects of chemicals used in hair care products….” “This might be one of the causes of the different cortisol levels seen in the hair of the male and female subjects in the current study.”

Did the authors collect any data that participants used hair dye when collecting their hair samples? If so, what percent of the female did dye their hair?  In this case/manuscript, do the male cortical levels in their hair more reflect the actual levels compared with female’s ?  If so, it should be discussed.

Response: Thank you for the raising a very important point for discussion. The authors agree with this point “exploring the effect of hair treatment/dye/coloring on the cortisol levels” of the worthy reviewer. The current study was designed with the goals of comparing and evaluating the levels of cortisol variation in saliva and hair samples taken from young adult dental students at two time intervals. And therefore, the effect of the hair treatment was not explored. However, the present team of the authors/researchers are working on the same topic for their future research studies and like some other valuable variables, this important variable i.e., the effect of hair treatment on the cortisol levels, will also be incorporated and investigated in the upcoming projects. For this manuscript, this is added as limitations and suggestions for future studies [Lines 292-294].

2, Line 266: “This pattern was also shown in the results of the current investigation, where the levels of cortisol in the saliva and hair samples of the male participants were higher than those of the female participants.”

But, Figure 3 and 4 show that there is no significantly differences of cortisol level in Saliva between male and female at 1st collection: Why?  Is there any significantly difference of those between male and female at 2nd collection?

Response: Thank you for your valuable suggestion and inquiry. In the paragraph that is referred by the worth reviewer, the point of higher cortisol levels among the male subjects as compared to the female subjects is discussed. In the figure 3 it can be clearly observed that the cortisol levels for the male subjects both at the 1st collection and the 2nd collection are higher as compared to the female subjects.

Finally, the authors would like to thank the reviewer for the constructive feedback and helping the authors to improve the contents and quality of this manuscript. We hope the quality of the manuscript has been improved and will be acceptable for publication.

Sincerely,

Reviewer 3 Report

Interestingly done and accurate research. A way to treat women's hair can be added. A study of cortisol in association with other hormones may be done in the future for greater completeness of the study.

Author Response

Thank you for the raising a very important point for discussion. The authors agree with this point “A study of cortisol in association with other hormones” of the worthy reviewer. The current study was designed with the goals of comparing and evaluating the levels of cortisol variation in saliva and hair samples taken from young adult dental students at two-time intervals. However, the present team of the authors/researchers are working on the same topic for their future research studies and like some other valuable variables, this important variable i.e., cortisol in association with other hormones , will also be incorporated and investigated in the upcoming projects. For this manuscript, this is added as limitations and suggestions for future studies [Lines 292-296].

Round 2

Reviewer 1 Report

Good answers - accept

Author Response

The authors would like to thank the respected reviewer for reviewing and providing an insightful and constructive feedback on the manuscript that certainly helped us to improve content. The positive feedback has greatly inspired us. Thank you.